# Isolation and Biochemical Characterization of Six Anaerobic Fungal Strains from Zoo Animal Feces

**DOI:** 10.3390/microorganisms9081655

**Published:** 2021-08-03

**Authors:** Marcus Stabel, Tabea Schweitzer, Karoline Haack, Pascal Gorenflo, Habibu Aliyu, Katrin Ochsenreither

**Affiliations:** Process Engineering in Life Sciences 2: Technical Biology, Karlsruhe Institute of Technology, 76131 Karlsruhe, Germany; tabea-schweitzer@web.de (T.S.); karoline_haack@hotmail.de (K.H.); pascal.gorenflo@kit.edu (P.G.); habibu.aliyu@kit.edu (H.A.); katrin.ochsenreither@kit.edu (K.O.)

**Keywords:** *Neocallimastigomycota*, lignocellulose, metabolites

## Abstract

Anaerobic fungi are prime candidates for the conversion of agricultural waste products to biofuels. Despite the increasing interest in these organisms, their growth requirements and metabolism remain largely unknown. The isolation of five strains of anaerobic fungi and their identification as *Neocallimastix cameroonii*, *Caecomyces spec.*, *Orpinomyces joyonii*, *Pecoramyces ruminantium*, and *Khoyollomyces ramosus*, is described. The phylogeny supports the reassignment *of Neocallimastix californiae* and *Neocallimastix lanati* to *Neocallimastix cameroonii* and points towards the redesignation of *Cyllamyces* as a species of *Caecomyces*. All isolated strains including strain A252, which was described previously as *Aestipascuomyces dubliciliberans*, were further grown on different carbon sources and the produced metabolites were analyzed; hydrogen, acetate, formate, lactate, and succinate were the main products. *Orpinomyces joyonii* was lacking succinate production and *Khoyollomyces ramosus* was not able to produce lactate under the studied conditions. The results further suggested a sequential production of metabolites with a preference for hydrogen, acetate, and formate. By comparing fungal growth on monosaccharides or on the straw, a higher hydrogen production was noticed on the latter. Possible reactions to elevated sugar concentrations by anaerobic fungi are discussed.

## 1. Introduction

The reduction of greenhouse gas emissions to limit global warming is one of the biggest challenges scientists need to address. Using renewables like lignocellulosic materials instead of fossil resources for the production of energy carriers and materials is one option to lower the CO_2_ footprint. However, the usage of agricultural waste products like straw as carbon source for microorganisms requires pretreatment to make carbohydrates accessible due to the recalcitrance of lignocellulose. These pretreatments are expensive, energy intensive and/or ineffective [1]. Additionally, toxic substances like furfural and 5-hydroxymethyl furfural are produced by some methods inhibiting microbial growth and product formation [2]. Anaerobic fungi (AF), which inhabit the gut of a great variety of herbivores [3], do not require pretreatment of lignocellulose to use it as substrate [4], making them excellent candidates for biofuel production. As demonstrated recently, they were used together with *Escherichia coli* in a two stage process to produce ethanol from corn stover [5]. Anaerobic fungi mediate the breakdown of ingested lignocellulose in their host, using a mix of mechanical disruption [6,7] and secretion of a big variety of hemicellulases and cellulases [8]. These enzymes often form multi enzyme complexes called cellulosomes [9,10]. Enzyme preparations from AF have shown similar activities as commercial formulations [11] which makes them excellent candidates for the screening of lignocellulose degrading enzymes. Despite their potential, AF seem to be lacking lignin degrading capabilities [8,12].

Historically, AF were first described as flagellated protozoans [13], which was contradicted by Orpin in 1975 [14] by demonstrating the occurrence of chitin in the cell walls of zoospores [15]. In 2007, anaerobic fungi were unified in a new phylum, *Neocallimastigomycota,* before being placed systematically with the closely related *Chytridiomycota* [16]. To the best of our knowledge, 20 genera of anaerobic fungi have been described until now: *Neocallimastix*, *Piromyces*, *Caecomyces*, *Cyllamyces*, *Anaeromyces*, *Orpinomyces*, *Oontomyces*, *Buwchfawromyces*, *Pecoramyces*, *Liebetanzomyces*, *Feramyces*, *Agrisomyces*, *Aklioshmyces*, *Capellomyces*, *Ghazallomyces*, *Joblinomyces*, *Khoyollomyces*, *Tahromyces*, *Aestipascuomyces*, and *Paucimyces* [17,18,19,20,21,22,23,24,25,26,27,28,29]. While in the past morphological features played a major role in describing AF [30], nowadays DNA barcoding plays a decisive role. Lately, the D1/D2-domain of the large ribosomal subunit (LSU) has been used as marker, being more consistent than the previously-used internal transcribed spacer region 1 (ITS1) [26,31].

Anaerobic fungi gain energy through mixed acid fermentation, with acetate, formate, succinate, lactate, ethanol, and hydrogen being the main products [32,33]. Production of hydrogen, acetate, and formate occurs mainly in the hydrogenosomes, organelles which have evolved from mitochondria lacking in AF [34]. Studies analyzing the metabolites produced during AF growth are scarce and are mainly focused on the genera *Piromyces* and *Neocallimastix*. With the exact nutritional needs of AF being unknown but highly relevant for the biotechnological use of these promising organisms [35], such studies are of high importance. To the best of our knowledge any study has so far compared the produced metabolites from several AF species during growth on multiple carbon sources. Here, the isolation of five strains of AF belonging to the genera *Neocallimastix*, *Orpinomyces*, *Caecomyces*, *Pecoramyces*, and *Khoyollomyces* is reported. The biochemical characterization of these isolates as well as of isolate A252, which was previously identified [20], was also performed.

## 2. Materials and Methods

### 2.1. Media

Enriched medium was adapted from the basal medium as previously described [26]. One liter contained 150 mL salt solution A, 150 mL salt solution B, 3 g yeast extract, 10 g tryptone, 2 mL hemin solution, and 2 mL resazurin solution. Distilled water was added to a total weight of 850 g, and the solution was boiled by microwaving to reduce the oxygen content, as indicated by the color change of resazurin to red. After readjusting the weight to 850 g with distilled water, the solution was cooled on ice, and 150 mL clarified rumen fluid, 6 g NaHCO_3_ and 1 g cysteine hydrochloride monohydrate were added. Subsequently, enriched medium was supplemented with the following carbon sources: dissolved xylan solution with 2 g dry weight/l, 2 g/L cellobiose and 5 g/L wheat straw (milled to a particle size of 1 mm with a Pulverisette 25, Fitsch, Idar-Oberstein, Germany). The medium was deoxygenated by gassing with 100% CO_2_ until turning yellow. Afterwards the pH was adjusted to 6.9 with 5M NaOH, and 50 mL of the medium was dispensed into serum bottles (118 mL) prior flushed with CO_2_. These were closed with a butyl stopper, fastened with an aluminum crimp seal, and subsequently sterilized by autoclaving at 121 °C for 20 min. The medium for the roll tube isolation technique was prepared identically except the addition of 0.9 g agar to each serum bottle and different carbon sources. Here, no straw and 3 g/L cellobiose instead of 2 g/L were used. The roll tubes were stored at 60 °C after autoclaving to avoid agar solidification. Before inoculation of any media, 0.5 mL of antibiotics solution was added to each serum bottle.

Salt solution A contained KH_2_PO_4_ (3.0 g/L), (NH4)_2_SO_4_ (6.0 g/L), NaCl (6.0 g/L), MgSO_4_ 7H_2_0 (0.6 g/L), and CaCl_2_ 2H_2_O (0.6 g/L) in distilled water. Salt solution B contained K_2_HPO_4_ (3.0 g/L) in distilled water. Hemin solution was prepared by dissolving 0.05 g hemin in 50 mL ethanol and then adding 50 mL of 0.05 M NaOH. The resazurin solution was made by dissolving 1 g/L resazurin in distilled water. Dissolved xylan was prepared according to [36] by adding 50 g of beechwood xylan (Carl Roth) to 1 L of distilled water, adjusting the pH value to 10 with 5M NaOH, and incubating at room temperature for 1 h. The solution was centrifuged for 10min at 10,000 ×*g*, the supernatant was transferred to a beaker, and the pH value was adjusted to 7 with 1 M acetic acid. The quantity of dissolved xylan was determined by freeze drying 10 mL of the solution and weighing the dried substance. The antibiotics solution contained 120 g/L of each penicillin sodium salt, ampicillin sodium salt, and streptomycin sulfate dissolved in distilled water, and was filter sterilized.

Defined medium was prepared similarly to enriched medium with the following changes: clarified rumen fluid was replaced by 140 mL distilled water, yeast extract and tryptone were omitted, 10 mL of the trace elements solution were added during medium deoxygenation, and 0.5 mL of the vitamin solution was added just before fungal inoculation. When using insoluble carbon sources (wheat straw, agarose, xylan, starch, inulin, chitin, pectin, alginate, or crystalline cellulose), 0.25 g of each substrate was added to a serum bottle flushed with 100% CO_2_ and subsequently, aliquots of 49.5 mL media were added to achieve a final concentration of 5 g/L of each carbon source. When using soluble carbon sources (cellobiose, maltose, trehalose, lactose, sucrose, glucose, xylose, mannose, fructose, arabinose, ribose, galactose, or glucuronic acid), the aliquots consisted of 44.5 mL media, and 5 mL carbon source were added from a 50 g/L stock to each serum bottle for a final concentration of 5 g/L before fungal inoculation. 

The vitamin solution was prepared in distilled water and contained 0.01 g/L thiamine, 0.2 g/L riboflavin, 0.6 g/L calcium pantothenate, 0.6 g/L nicotinic acid, 1.0 g/L nicotinamide, 0.05 g/L folic acid, 0.02 g/L vitamine B12, 0.2 g/L biotin, 0.1 g/L pyridoxamine, and 0.05 g/L p-aminobenzoic acid [37] and was filter sterilized after preparation. The trace elements solution was prepared according to Lowe et al. 1985 [38] and contained 0.25 g/L MnCl_2_ 4H_2_O, 0.25 g/L NiCl_2_ 6H_2_O, 0.25 g/L NaMoO_4_ 2H_2_O, 0.25 g/L H_3_BO_3_, 0.20 g/L FeSO_4_ 7H_2_O, 0.05 g/L CoCl_2_ 6H_2_O, 0.05 g/L Na_2_SeO_3_ 5H_2_O, 0.05 g/L NaVO_3_ 4H_2_O, 0.025 g/L ZnSO_4_, 0.025 g/L CuSO_4_ 2H_2_O in 0.2 M HCl.

### 2.2. Isolation of AF

Feces from reticulated giraffe (*Giraffa reticulata* “G”), watussi cattle (*Bos primigenius f. taurus,* “W”), Przewalski’s horse (*Equus ferus,* “PP”), and scimitar oryx (*Oryx dammah,* “SA”) were collected by personal from the Karlsruhe Zoo (Germany) in the afternoon (8 April 2019) and stored at room temperature in plastic sample beakers until the next day. Feces of horse (*Equus caballus,* “X”) was collected in a barn in Reilingen, Baden Württemberg, Germany (25 January 2019) and used directly for isolation as described below. A 1:1 mixture of salt solution A and salt solution B was deoxygenated by gassing with 100% CO_2_ for 30 min. One gram of feces was added to the solution and dispersed by further gassing. Enriched medium was inoculated with 5 mL of this suspension and incubated for 7 days at 39 °C in darkness in a BBD 6220 incubator (Heraeus Deutschland GmbH & Co., Hanau, Germany). Inoculations were carried out by transferring 5 mL of culture with a syringe to the recipient vessel. After the incubation, fungal growth was monitored by light microscopy. Serum bottles with visible anaerobic fungal growth were chosen to inoculate roll tubes which had been prepared as described above. After inoculation, the bottle was placed on ice and rolled to achieve a fine layer of solidified medium on the inner surface of the serum bottle. The roll tubes were incubated for 4 days at 39 °C in darkness. By picking different colony types from the roll tubes, various strains of anaerobic fungi were separated from other microorganisms. Picked colonies were transferred to fresh enriched medium and incubated for another 7 days followed by another passage of roll-tubing. After 3 passages of roll-tubing and observation by light microscopy, all cultures were considered as pure fungal isolates.

### 2.3. Identification of AF

Genomic DNA was isolated from the entire mycelium of one-week old cultures grown in 50 mL defined medium with cellobiose as C-source using the Quick-DNA Fecal/Soil Microbe DNA Miniprep Kit (Zymo Research, Orange County, CA, USA) following the instructions of the manufacturer. The amount of isolated gDNA was determined using a microplate spectrophotometer (Epoch, BioTek Instruments GmbH, Bad Friedrichshall, Germany). The part of the ribosomal operon between the 18S small ribosomal subunit and the D1/D2 region of the 28S large ribosomal subunit was amplified as described by Wang et al. (2017) [39] using the ITS5 [40] and NL4 [41] primers (NL4: GGTCCGTGTTTCAAGACGG; ITS5: GGAAGTAAAAGTCGTAACAAGG). The PCR reaction was composed of 12.5 µL of Q5 High-Fidelity 2X Master Mix (New England Biolabs (NEB), Ipswich, MA, USA), 0.5 µL of each primer (10 µM stock) and 20 ng gDNA in 25 µL reaction volume. The PCR program consisted of 30 s initial denaturation at 98 °C, 30 cycles of 10 s denaturation at 98 °C, 30 s annealing at 62 °C, 90 s elongation at 72 °C and a final extension step at 72 °C for 2 min. The resulting 1.5 kb fragments were separated in a 2% agarose gel run at 120 V and stained with ROTI^®^GelStain (Carl Roth, Karlsruhe, Germany). Bands were cut out from the gel and the DNA was eluted with a Monarch Gel Extraction Kit (NEB). Recovered DNA fragments were cloned using a PCR Cloning Kit (NEB) and transformed into NEB 10-beta *Escherichia coli* following the supplier’s instructions. Transformants were selected by streaking on LB-agar containing 10 g/L NaCl, 10 g/L trypton, 5 g/L yeast extract, 20 g/L agar, and 0.05 g/L ampicillin sodium salt. The agar-plates were incubated overnight at 37 °C. Colonies were picked and grown overnight in LB-media as described above (no agar) at 37 °C under shaking at 210 rpm in a Multitron Shaker (Infors, Bottmingen, Switzerland). Plasmids were extracted using the Monarch Plasmid Miniprep Kit (NEB) and sequenced by the Supreme Run Sanger Sequencing Service from Eurofins Genomics using the sequencing primer from the PCR Cloning Kit. The obtained sequences were uploaded to genbank with following accession numbers: Isolate G341 MW175298-175304, Isolate W212 MW175312-175318, Isolate SA222 MW175305-MW175311, Isolate X2152 MW175319-MW175325 and Isolate PP313 MW175289-175297. The ribosomal operon between the 18S small ribosomal subunit and the D1/D2 region of the 28S large ribosomal subunit for *Neocallimastix californiae*, *Neocallimastix lanati*, and *Caecomyces churrovis* was extracted from the respective published genomes [33,42,43] using BioEdit V. 7.0.5.3 [44]. For the LSU analysis, a database with *Neocallimastigomycota* sequences, the sequences of our isolates and the sequences extracted from the genomes were used encompassing a total of 244 sequences. *Gonopodya prolifera* (JN874506) was included as outgroup. Phylogenetic analysis was performed by aligning the sequences from the isolates with the databases using MAFFT v7.471 [45] at https://mafft.cbrc.jp/alignment/server/ (Accessed on 10 June 2021). The resultant alignments were manually curated and trimmed using BioEdit. Phylogenies were reconstructed using IQ-TREE v2.0.3 [46] with model selection [47] and 1000 ultrafast bootstraps [48]. The obtained trees were visualized and edited with Evolview V.3 [49]. Sequence identity matrices were calculated using BioEdit.

### 2.4. Carbon Source Usage and Analytics

Five mL of culture liquid from the isolates and *Aestipascuomyces dubliciliberans* strain A252 [20] were inoculated into defined media prepared as described above with 5 g/L of one of the following carbon sources: wheat straw, xylan, starch, inulin, chitin, pectin, alginate, crystalline cellulose, cellobiose, maltose, trehalose, lactose, sucrose, glucose, xylose, mannose, fructose, arabinose, ribose, galactose, or glucuronic acid. Incubation followed at 39 °C in the dark. After five to seven days, they were transferred into fresh defined media. After three passages, biomass formation was evaluated by eye, light microscopy, and pressure measurement through a manometer. Cultures with visible growth were inoculated into a fourth passage of seven days growth in triplicate. The composition of the liquid phase was measured by HPLC by taking 1 mL samples directly after inoculation and on the seventh day. Samples were prepared for HPLC by centrifuging at 10,000 g for 10 s and filtering the supernatant with 0.22 µm cellulose acetate syringe filters (Restek). Subsequently, 10 µL of supernatant were injected to a Rezex ROAorganic acid H + (8%) column from Phenomex in a 1100 Series System from Agilent Technologies with 5 mM sulfuric acid as eluent at a flow rate of 0.5 mL/min and a column temperature of 50 °C. Sugars and ethanol were detected with a refractive index detector and organic acids by a diode array detector at 220 nm. The gaseous phase was analyzed by GC by taking 5 mL of sample from the gaseous phase with a syringe after measuring the pressure on the fifth day. This sample was injected manually into a 3000 Micro GC System from Inficon with a CP-Molsieve 5 Å column and a PoraPLOT Q column at 80 °C.

## 3. Results

### 3.1. Isolation and Phylogeny

From each host animal a single fungal isolate was obtained: G341 from giraffe (*Giraffa reticulata*), PP313 from Przewalski’s horse (*Equus ferus)*, W212 from watussi cattle (*Bos primigenius f. taurus*), SA222 from scimitar oryx (*Oryx dammah*), and X2152 from horse (*Equus caballus*). The LSU phylogeny was inferred in IQtree based on 245 sequences (alignment: 691 characters) from the above isolates (G341, PP313, W212, SA222, and X2152), and other AF using *Gonapodya prolifera* strain ATCC MYA-4800 as outgroup. All isolates could be assigned at least on genus level to previously described AF genera (Figure 1) in well-supported monophyletic groups. The sequences from G341 grouped into the sequences assigned to *Neocallimastix cameroonii*, W212 into sequences of *Orpinomyces joyoonii*, SA222 into the ones from *Pecoramyces ruminantium* and X2152 into sequences of *Khoyollomyces ramosus*. According to our phylogeny, the sequences from the genomes of *Neocallimastix californiae* and *Neocallimastix lanati* grouped inside the *Neocallimastix cameroonii* clade. The identity of the aligned and trimmed sequences to the assigned species were 99.0–100% for G341, 99.0–99.6% for W212, 97.8–100% for SA222, and 98.9–99.6% for X2152 (Appendix A), respectively. The genomic sequences from *Neocallimastix californiae* shared 99.6% and the ones from *Neocallimastix lanati* shared 99.5–100% identity with the sequences from *Neocallimastix cameroonii*.

However, the affiliation by LSU phylogeny of the isolate PP313 was less clear. The PP313 sequences grouped into a clade consisting of sequences from *Caecomyces communis*, *Cyllamyces aberensis* and *Caecomyces churrovis*. The sequences of PP313 and those extracted from the *Caecomyces churrovis* genome cluster distinctly in two subclades. PP313 shared 98.7–99.0% sequence identity with *Caecomyces communis*, 98.4–99.8% with *Caecomyces churrovis* and 96.7–98.7% with *Cyllamyces aberensis*. Interestingly, the sequences from *Cyllamyces aberensis* grouped inside the *Caecomyces* sequences forming a group with sequences from *Caecomyces communis* and sharing 97.5–98.6% sequence identity. The sequence identity between *Caecomyces churrovis* and *Cyllamyces aberensis* was 98.1–98.7%.

As for the whole bulbous clade, sequences of the D1/D2 region were either highly similar and/or LSU sequences were not available. ITS1 region was therefore used to study the clade. In the ITS phylogeny sequences from PP313 were placed together with published (MF460993) and genome extracted sequences from *Caecomyces churrovis* (Appendix A). In the corresponding identity matrix, the sequences from the genome shared 98.3–100% identity with MF460993 (Appendix A). PP313 shared 99.1% identity with both MF460993 and the genomic sequences. *Cyllamyces aberrensis* grouped outside the *Caecomyces* clade in this ITS1 phylogeny (Appendix A).

### 3.2. Carbon Source Usage

The substrate utilization range by isolates, as well as the *Aestipascuomyces dubliciliberans* strain A252, was analyzed and evaluated by quantification of the produced metabolites. Ethanol, while being detected during the growth of all strains, could not be quantified reliably due to sample preparation which facilitated evaporation. Therefore, ethanol amounts are not detailed further. Furthermore, if the total amount of produced metabolites was lower than 1 mmol in the fourth passage, growth was considered as low. The total amount of produced metabolites on the different C-sources excluding ethanol is given in Table 1. None of the fungi was able to grow on chitin, alginate, arabinose, ribose, galactose, and glucuronic acid.

G341 grew on wheat straw, xylan, starch, inulin, pectin, cellulose, cellobiose, maltose, lactose, sucrose, glucose, xylose, mannose, and fructose. The isolate showed low growth on trehalose (0.008 ± 0.003 mmol).

PP313 grew on wheat straw, xylan, cellobiose, lactose, glucose, xylose, and fructose. Low growth was observed on starch (0.204 ± 0.152 mmol), pectin (0.352 ± 0.155 mmol), cellulose (0.093 ± 0.047 mmol), and maltose (0.080 ± 0.038 mmol).

W212 utilized wheat straw, xylan, starch, cellulose, cellobiose, maltose, lactose, sucrose, glucose, xylose, and fructose while showing low growth on pectin (0.269 ± 0.085 mmol).

SA222 could use wheat straw, xylan, starch, cellulose, cellobiose, maltose, lactose, sucrose, glucose, xylose, mannose, and fructose. Low growth was observed on pectin (0.393 ± 0.027 mmol) and trehalose (0.005 ± 0.000 mmol).

X2152 grew on wheat straw, xylan, cellulose, cellobiose, lactose, glucose, xylose, and fructose. In addition, low usage of starch (0.008 ± 0.001 mmol), pectin (0.408 ± 0.063 mmol), maltose (0.035 ± 0.002 mmol), trehalose (0.006 ± 0.000 mmol), and sucrose (0.246 ± 0.042 mmol) was observed.

A252 grew on wheat straw, xylan, starch, cellulose, cellobiose, maltose, lactose, sucrose, glucose, xylose, mannose, and fructose. Low growth was observed on inulin (0.315 ± 0.419 mmol), pectin (0.444 ± 0.150 mmol) and trehalose (0.007 ± 0.005 mmol).

As the number of detected metabolites on trehalose was low for all isolates, it might be considered as an experimental artefact and will not be discussed further. The overall highest cumulated metabolite concentration was detected for A252 growing on cellobiose (4.434 ± 0.070 mmol), but it also consistently produced the highest metabolite concentrations with most of the C-sources (Table 1) except when growing on cellulose (W212) and inulin or xylose which were best utilized by G341.

Table 2 displays the carbon balance excluding the not quantifiable ethanol for the soluble C-sources. For PP313, the number of produced metabolites on maltose was so low and variant that a precise calculation of the carbon balance was not conclusive. The total carbon balance was calculated from the samples of cellobiose, fructose, glucose, maltose, mannose, and xylose. When one of the C-sources was not usable by an isolate, it was omitted in the calculation. On lactose, the carbon balance was seemingly much lower for all isolates than on other carbon sources except for strain A252. It was observed that most strains consumed preferably the released glucose from lactose but the residual galactose only to a much lower extent. When the residual galactose was added to the product side of the carbon balance relatively high values were achieved compared to the average carbon balance (Table 2). The amount of residual galactose varied highly between isolates (Appendix A) with the lowest amount being 0.043 ± 0.008 mmol for A252, but for all tested fungi a small decrease in galactose was determined while growing on lactose.

### 3.3. Metabolite Production

Figure 2 shows the total amount of metabolites in mmol, including their relative distribution. The main detected metabolites were hydrogen, acetate, and formate, which were produced by all isolates. Lactate production was observed for all fungi but X2152, where it was only detected in one of three samples of sucrose cultures. Succinate was produced by all isolates but W212, while for SA222 its production was lower compared to the other isolates and restricted to fructose, glucose, starch, sucrose, and xylan cultures. Citrate was only detected in traces in samples of xylan, straw, or pectin cultures of strain SA222. Next, the dependence of the metabolite composition from the C-source was analyzed. Therefore, the number of single metabolites was compared to the total amount of produced metabolites (Figure 3). Although no big changes were apparent between C-sources that yielded in roughly the same total amount of metabolites, prominent differences were noticeable when compared to C-sources which only yielded in a small total amount of metabolites. Here, less metabolite diversity was observed. As the total amount increased, not only did the diversity of metabolites increase, but the metabolites seemed to be produced in a specific order. Hydrogen was always present with formate and acetate appearing with increasing total metabolite amount followed by lactate. Strain PP313 was an exception as it produced lactate subsequent to hydrogen, then acetate and finally formate. Succinate seems to be produced only when an abundant production of other metabolites occurred (Figure 2), although it remained a marginal metabolite (Figure 3). Another factor influencing the composition of the metabolites seemed to be the complexity of the C-source, as the most complex C-source (wheat straw) resulted in a high relative (Figure 2) as well as absolute (Figure 3) amount of produced hydrogen. The analysis below will focus on straw and the monosaccharides fructose, glucose, and xylose as these enabled good growth for all tested fungi. Both absolute and relative values for all strains on all C-sources are given in the Appendix A.

G341 produced 0.352 mmol of hydrogen while growing on straw followed by 0.312 mmol on glucose, 0.232 mmol on fructose, and 0.278 mmol on xylose, although the total amount of metabolites being 2.290 mmol on straw compared to 3.736–3.925 mmol for the monosaccharides. In relation to the total amount of metabolites, 15.4% hydrogen were produced on straw in comparison to 6.1–8.4% on monomeric sugars. Strain G341 grown on straw also produced lower portions of lactate (6.1%) and succinate (ND) compared to growth on xylose, glucose, or fructose (18.2–19.7% lactate and 3.0–3.4% succinate). For formate, concentrations were slightly higher in straw cultures (44.4%) than in cultures using monomeric sugars (37.0–39.1%), whereas acetate concentrations were comparable (33.6% vs. 31.4–34.7%).

Isolate PP313 produced less hydrogen when grown on straw (0.322 mmol) than on the monosaccharides (0.400–0.488 mmol). Although in relation to the total amount of metabolites hydrogen was more elevated on straw than on monosaccharides (20.6% vs. 12.4–15.5%), the differences between the C-sources were not as pronounced as for the other isolates and might be explained by the incrementing diversity explained above.

Metabolite production by strain W212 and G341 followed a similar pattern. Hydrogen increased significantly while growing on straw (0.375 mmol) in comparison to monosaccharides (0.097–0.194 mmol) despite a lesser number of total metabolites (2.312 mmol vs. 3.271–3.608 mmol) being produced. On straw lactate amounted to 6.2%, while acetate and formate yielded in 39.3% and 37.7% of the total metabolites, respectively. In comparison, cultivation with monosaccharides resulted in increased formate (48.3–56.8%) and decreased acetate (29.0–33.34%) levels. The relative lactate amount spanned a range between 6.4% (glucose) and 19.7% (xylose) and seemed to be increasing with the decreasing relative amount of formate. As described above, succinate could not be detected for W212.

Strain SA222 also produced the highest amount of hydrogen when growing on straw (0.303 mmol) compared to the monosaccharides (0.193–0.260 mmol) despite the differences in total metabolite amount (1.709 mmol for straw vs. 3.286–3.399 mmol for monosaccharides). While the portion of acetate when growing on straw (35.7%) was higher compared to the monosaccharides (25.3–27.8%), the portion of lactate was lower (3.4% on straw vs. 17.6–25.6% on monosaccharides). While growing on straw and xylose, succinate was not detected, and it amounted to 0.9% on glucose and 2.0% on fructose. When comparing the monosaccharides, a relation between a higher percentage of succinate (2.0% for fructose vs. ND for xylose) and a lower percentage of lactate (17.6% on fructose vs. 25.6% on xylose) as well as hydrogen (5.9% on fructose vs. 7.8% on xylose) was observed. No correlations were found for the relative amount of formate which varied independently from the complexity of the C-source between 40.8% and 46.7%.

Isolate X2152 produced 0.411 mmol of hydrogen while growing on straw which was comparatively high in relation to 0.236–0.321 mmol for growth on monosaccharides, although the total amount of metabolites was considerably higher on sugars (3.5–3.9% vs. 1.7%). On straw, the relative amount of formate was slightly higher (53.8%) and the relative amount of acetate was lower (21.4%) when compared to growth on monosaccharides (49.3–50.6% formate and 35.1–36.7 acetate). No succinate was detected on straw, whereas it ranged between 5.3% and 8.3% for growth with monosaccharides. Lactate was not detected in any of the samples, as described above.

For strain A252, the highest total amounts of hydrogen (0.543 mmol) were observed in straw cultures, although total amount of metabolites (2.682 mmol) was relatively low when compared to growth on monosaccharides (0.430–0.500 mmol hydrogen of 3.664–4.039 mmol total metabolites). The relative amount of acetate was higher on straw (39.3%) compared to the growth with monosaccharides (29.4–31.0%). All other metabolites were produced in lower portions on straw (33.6% formate, 3.2% lactate, 3.1% succinate) than when growing on monosaccharides (37.7–38.6% formate, 14.2–15.9% lactate, 4.2–5.1% succinate).

In all samples with straw as C-source a small amount of citrate was detected, ranging between 0.2% (SA222) to 0.6% (X2152) of the total metabolites.

## 4. Discussion

### 4.1. Phylogeny

In this study five different fungal species of *Neocallimastigomycota* from fecal sample of different zoo animals were isolated. Four of the isolates could be assigned to known anaerobic fungal species: isolate G341 to *Neocallimastix cameroonii*, isolate W212 to *Orpinomyces joyoonii*, isolate SA222 to *Pecoramyces ruminantium*, and isolate X2152 to *Khoyollomyces ramosus*. Our phylogeny showed clustering of isolate G341 together with *Neocallimastix californiae* and *Neocallimastix lanati* and *Neocallimastix cameroonii* in a well-supported clade and shared high sequence identity, indicating that *Neocallimastix californiae*, *N.*
*lanati*, and *N.*
*cameroonii* may be conspecific. The description of *Neocallimastix cameroonii* [50] preceded *N. californiae* [51] and *N.*
*lanati* [33] and hence should have taxonomic preference as the species name. This would reduce the number of described *Neocallimastix* species to two: *Neocallimastix frontalis* and *Neocallimastix cameroonii*. The *Neocallimastix* species *N. patriciarum*, *N. variabilis* and *N. hurleyensis* [52,53,54] described in the past were later also shown to be part of the already described species *N. frontalis* [30,39]. This highlights the need for a more robust phylogenetic framework that may include multiple genetic markers or whole genome sequences to resolve the phylogeny of anaerobic fungi.

Isolate PP313 clustered inside a clade of sequences from both *Caecomyces* and *Cyllamyces* in the LSU phylogeny. While sharing the highest sequence identity with *Caecomyces churrovis* the sequences of PP313 clustered separately as an own clade. In contrast, when using the ITS1 region as a phylogenetic marker PP313 clustered together with sequences of *Caecomyces churrovis*. The genera of *Caecomyces* and *Cyllamyces* are the only anaerobic fungi with bulbous growth pattern compared to the other genera which are forming rhizoids [22,55]. The two genera *Caecomyces* and *Cyllamyces* are thought to be closely related and their taxonomic status has been the subject of debate. For instance, Wang et al. previously recommended more LSU sequencing to resolve the affiliations of the two genera [39]. To the best of our knowledge no phylogeny of the D1/D2 LSU region, including *Caecomyces churrovis*, has been reported so far. According to previous reports [26,39], *Cyllamyces* grouped in a clade together with different *Caecomyces* species in our LSU phylogeny. Conversely, but in accordance to previous reports [19,20,56], *Cyllamyces* grouped outside of the *Caecomyces* sequences in the ITS1 phylogeny. The ITS1 region is highly divergent even within the same strain and *Caecomyces* and *Cyllamyces* have an especially high sequence divergence in the ITS1 region [57]. The lower heterogeneity of the D1/D2 region and the respective placement of *Cyllamyces* in the corresponding phylogeny, we suggest the reassignment of *Cyllamyces* as a species of *Caecomyces*. Because of the discrepancy between ITS and LSU phylogenies and the questions raised above concerning the *Caecomyces* clade, we assign PP313 to the genus *Caecomyces* but avoid affiliating to a species. The inclusion of additional taxonomic markers could help resolve the highlighted discrepancies.

Interestingly, only one fungal strain was isolated from each sample. It has been shown before that the frequency of isolation of an AF strain correlates with its abundance in the sample [57]. The same study reports that special effort was invested in obtaining several strains from the same sample. As the objective of this study was to isolate more robust species for a possible future biotechnological application, no special effort was undertaken for isolating as many strains as possible from each sample in contrast to the aforementioned study. Therefore, it is highly probable that only the most abundant or most robust species of each sample was isolated.

### 4.2. Carbon Source Usage

Of all isolates, strain G341 used the widest substrate range. Whereas no information concerning the substrate usage of the *Neocallimastix cameroonii* type strain was available, N. *californiae* was reported to be able to grow on glucose, cellobiose, cellulose, fructose, maltose, and different grasses [58,59]. The most extensive analysis on the carbon source usage has been performed recently for N. *lanati* [33]. In accordance with these experiments, isolate G341 was able to grow on glucose, cellobiose, fructose, maltose, cellulose, and xylan. Wilken et al. also used a model to predict further usable carbon sources, i.e., galactose, mannose, and xylose, which could not be confirmed in growth experiments [33]. In contrast, G341 was able to grow on most of these substrates confirming the model prediction. The difference between the two experiments is probably due to variances in media or experimental handling. The exception was galactose. While G341 did not grow on pure galactose, it seemed to be partly co-consumed when growing on lactose. Interestingly, N. *californiae* has been reported to lack the required genes for galactose usage [59].

Of all tested isolates, substrate usage range of strain PP313 was the narrowest. In accordance to literature, it was able to grow on all carbon sources reported previously for *Caecomyces churrovis* [60] including cellobiose, glucose, fructose, and xylan. *Caecomyces churrovis* was also reported to grow on different types of grass. While we did not test grasses, PP313 was able to grow on wheat straw, affirming the ability to grow on complex lignocellulosic substrates. While in accordance with *Caecomyces churrovis* not being able to utilize arabinose, galactose, and mannose, PP313 was able to grow on lactose. In lactose cultures, a slight consumption of galactose could be measured. As reported and discussed previously for *Caecomyces churrovis* [60], PP313 showed a preference for soluble carbon sources and growth on cellulose was low.

The metabolic information about *Orpinomyces joyonii* is particularly scarce as only one publication was found describing it [61]. Similar to isolate W212, *O. joyonii* was reported to grow on cellulose, xylan, starch, glucose, cellobiose, fructose, maltose, lactose, and shows only poor growth on pectin. Interestingly, growth on galactose was also reported. In our case, strain W212 was only able to co-utilize galactose when it was applied as component of lactose, and even then it consumed the lowest amount of all tested fungi.

Isolate SA222 differed in the carbon source usage from *Pecoramyces ruminantium* [27]. Contrary to previous reports, SA222 was not able to grow on inulin and barely survived on trehalose. On the contrary, it was able to utilize both lactose and pectin which was not reported before. In accordance with previous publications, SA222 grew on xylan, starch, cellulose, cellobiose, maltose, sucrose, glucose, xylose, mannose, and fructose. Although previously no growth on wheat straw was reported, growth on different lignocellulose containing materials has extensively been studied.

Up to date and to the best of our knowledge, information about the substrate usage of *Khoyollomyces ramosus* has not been published yet, as both genus and species were just described recently [19]. This renders our characterization of isolate X2152 the first report in this field.

Strain A252 was able to grow on cellulose, mannose, and pectin in addition to the C-sources reported previously [20], probably due to the tenfold increase of C-source concentration. While the other *Aestipascuomyces dubliciliberans* isolate R4 was also able to grow on glucuronic acid, this still was not the case for A252. Similar to the other tested fungi, A252 was not able to grow on galactose, but when grown on lactose it metabolized nearly all of it, even leading to a similar C-balance as on other C-sources with good growth.

### 4.3. Metabolite Production

The main fermentation products reported up to date for anaerobic fungi are hydrogen, formate, acetate, ethanol, lactate, and succinate [32], although production of citrate and malate has also been reported [62,63]. Our study confirmed these findings, although some of the tested fungal isolates were only producing some of the mentioned metabolites, i.e., only traces of citrate and no malate were detected. *Orpinomyces joyonii* isolate W212, for example, did not produce any succinate, confirming previous studies in which the production of formate, acetate, lactate, ethanol, and hydrogen was reported [61]. Similarly, *Khoyollomyces ramosus* isolate X2152 did not produce lactate, while this was a common metabolite for the other tested strains. However, generally only scarce information is available about metabolic data for anaerobic fungi. Available reports focus on the genera *Piromyces* and *Neocallimastix* and little is known of other anaerobic fungi. When comparing metabolite production of different fungi growing on different C-sources yielding either in high or low total metabolite production we noticed a sequential appearance of different metabolites with increasing total metabolite concentration. Although we just sampled at the beginning and at the end of the experiment, we can speculate about a preference for certain end products: first hydrogen, then acetate and formate, lactate, and finally succinate. A possible exception from this rule could be PP313 for which lactate was detected in samples lacking formate and acetate. Future studies should address this hypothesis with more sampling points between start and end. As it is possible that low total metabolite concentrations on some carbon sources are due to delayed fungal growth, elongating the duration of the experiments could also be considered.

Another difference in the fermentation pattern appears when comparing the growth on wheat straw to the growth on different monosaccharides. Despite the total sum of produced metabolites being much higher during monosaccharide growth, the amount of hydrogen was higher when growing on straw. When looking on the relative parts of other metabolites, they varied greatly between growth on monosaccharides and straw with the exact pattern depending on the observed species. An exception was PP313, which showed a strong general preference for soluble C-sources as mentioned above. Past studies showed a repression of biomass degrading enzymes by free sugars [64], indicating a reaction of anaerobic fungi towards these conditions. Anaerobic fungi have been reported to contain both pyruvate formate lyase (PFL) and pyruvate ferredoxin oxidoreductase (PFO) [32,65]. Recently it was shown that while both enzymes are present, the main metabolic flux is carried by PFL [33]. The authors also speculated that during growth on high sugar concentrations PFL could be used and under more complex conditions, like lignocellulose, it could switch to PFO. A higher PFO flux would lead to a higher hydrogen concentration [33], which would confirm our findings of higher hydrogen concentrations when growing on straw. If the flux through PFO increases, the relative amount of produced formate should decrease. This could only be observed for the strains A252, W212 and X2152 whereas the relative concentrations were comparable for SA222 and even slightly increased for G341. A common reaction to high sugar concentrations is the effect known as overflow metabolism, Crabtree effect or Warburg effect, known from a multitude of organisms including *Escherichia coli* [66], *Lactobacillus plantarum* [67], *Bacillus subtilis* [68], *Saccharomyces cerevisiae* [69], and mammalian cells [69]. Under high sugar/fast growth conditions these organisms switch from energy efficient respiration to energy inefficient fermentation. Despite the oxygen dependent pathways not being present in anaerobic fungi, a proton gradient has been observed in the hydrogenosomes through which an ATPase could work [33]. A shift from this hydrogenosomal pathway, analogous to the respiratory chain, towards the cytosolic mixed acid fermentation in response to high sugar concentrations similar to an overflow metabolism could be possible. A reason for the discrepancy between the amounts of produced hydrogen independent of C-source type could also be the reabsorption of the produced hydrogen into the metabolism at later fermentation stages. Past studies showed the presence of a hydrogen dehydrogenase that under physiological conditions would catalyze the reaction from NAD(P)^+^ and H_2_ towards H^+^ and NAD(P)H [33]. If this enzyme would be expressed only at later growth stages, it could explain the relative decrease in hydrogen at higher total metabolite concentrations.

## Figures and Tables

**Figure 1 microorganisms-09-01655-f001:**
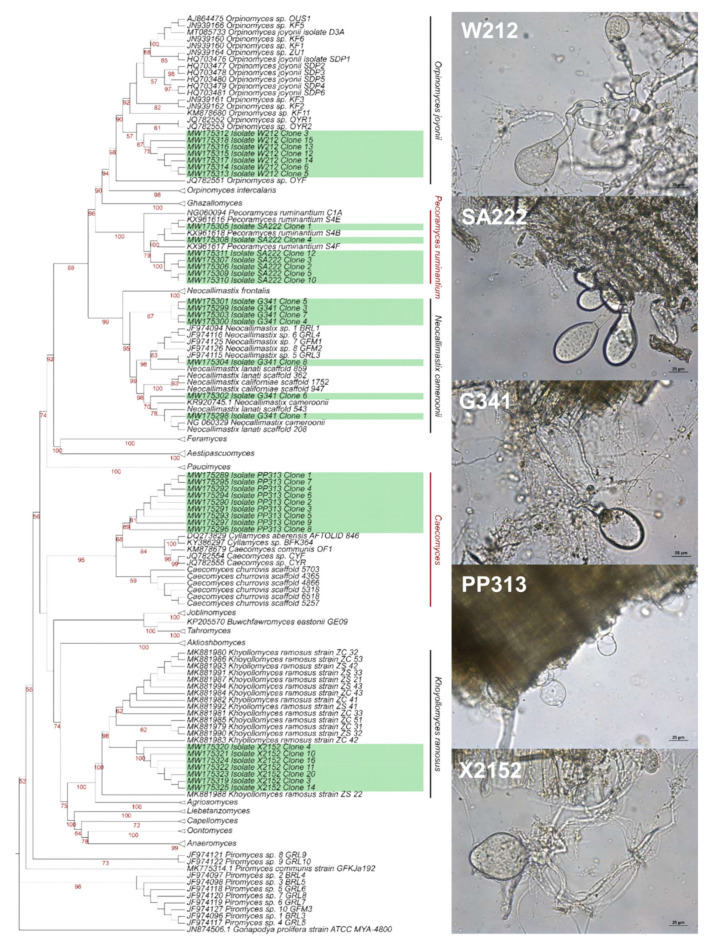
Left: Phylogenetic affiliation of the isolated strains to other AF genera based on the nucleotide sequences of the D1–D2 domains of the LSU from the ribosomal operon. MAFFT [45] was used for alignment and BioEdit [44] was used for manual curation of the sequences. The ML-tree was constructed using IQTREE [46] with the predicted model TN + F + R2 and –bb 1000. Bootstrap values higher than 50% are shown at the nodes. Right: Isolates growing in defined media with straw as c-source.

**Figure 2 microorganisms-09-01655-f002:**
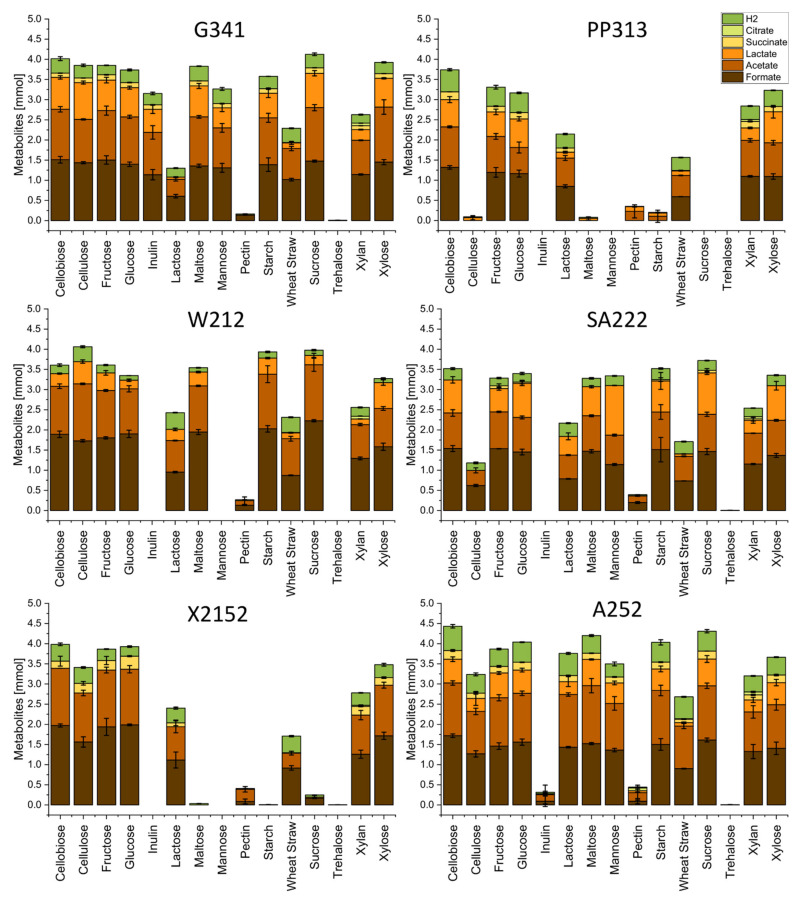
Produced metabolites (mmol) of the strains G341, PP313, W212, SA222, X2152 and A252 during growth on different C-sources. Ethanol is excluded. Values shown represent averages of triplicate experiments and standard deviations.

**Figure 3 microorganisms-09-01655-f003:**
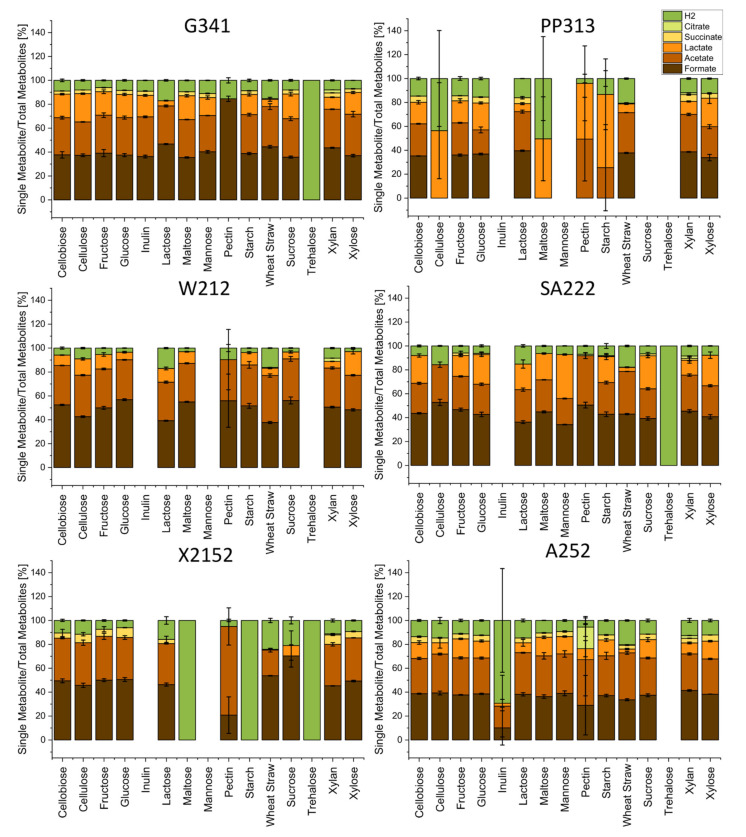
Relative amount of each produced metabolite when compared to the total of produced metabolites (%) of the strains G341, PP313, W212, SA222, X2152, and A252 during growth on different C-sources. Ethanol is excluded. Values shown represent averages of triplicate experiments and standard deviations.

**Table 1 microorganisms-09-01655-t001:** Total metabolites (mmol) without ethanol produced by the isolates during growth on different C-sources.

	Total Metabolites [mmol]					
	G341	PP313	W212	SA222	X2152	A252
Cellobiose	4.015 ± 0.079	3.741 ± 0.12	3.606 ± 0.154	3.519 ± 0.218	3.983 ± 0.052	4.434 ± 0.07
Cellulose	3.85 ± 0.069	0.093 ± 0.047	4.061 ± 0.015	1.181 ± 0.112	3.409 ± 0.159	3.238 ± 0.321
Fructose	3.85 ± 0.15	3.31 ± 0.247	3.608 ± 0.043	3.286 ± 0.09	3.865 ± 0.361	3.868 ± 0.184
Glucose	3.736 ± 0.02	3.167 ± 0.287	3.347 ± 0.192	3.399 ± 0.087	3.926 ± 0.085	4.039 ± 0.14
Inulin	3.153 ± 0.436	X	X	X	X	0.315 ± 0.419
Lactose	1.3 ± 0.104	2.146 ± 0.122	2.428 ± 0.035	2.168 ± 0.055	2.403 ± 0.418	3.759 ± 0.15
Maltose	3.83 ± 0.132	0.08 ± 0.038	3.542 ± 0.084	3.279 ± 0.087	0.035 ± 0.002	4.201 ± 0.187
Mannose	3.265 ± 0.35	X	X	3.339 ± 0.058	X	3.498 ± 0.268
Pectin	0.16 ± 0.007	0.352 ± 0.155	0.269 ± 0.085	0.393 ± 0.027	0.408 ± 0.063	0.444 ± 0.15
Starch	3.578 ± 0.397	0.204 ± 0.152	3.934 ± 0.293	3.521 ± 0.695	0.008 ± 0.001	4.032 ± 0.33
Wheat straw	2.29 ± 0.024	1.565 ± 0.021	2.312 ± 0.071	1.709 ± 0.025	1.707 ± 0.096	2.682 ± 0.08
Sucrose	4.125 ± 0.077	X	3.978 ± 0.232	3.721 ± 0.066	0.246 ± 0.042	4.309 ± 0.134
Trehalose	0.008 ± 0.003	X	X	0.005 ± 0	0.006 ± 0	0.007 ± 0.005
Xylan	2.627 ± 0.02	2.841 ± 0.061	2.557 ± 0.04	2.54 ± 0.101	2.781 ± 0.226	3.201 ± 0.415
Xylose	3.925 ± 0.273	3.23 ± 0.061	3.271 ± 0.156	3.357 ± 0.108	3.478 ± 0.209	3.664 ± 0.397

**Table 2 microorganisms-09-01655-t002:** Calculated carbon balance without ethanol of the isolates during growth on the C-sources cellobiose, fructose, glucose, maltose, mannose, and xylose. Ø is the average of these sugars. “Lactose” stands for the C-balance during growth on lactose with regard to the hydrolyzed amount of lactose, but not taking into account the actual consumption of released monosaccharides. “Lactose*” is the C-balance including residual galactose.

	C-Balance					
	G341	PP313	W212	SA222	X2152	A252
Cellobiose	0.787 ± 0.026	0.702 ± 0.035	0.597 ± 0.027	0.659 ± 0.050	0.633 ± 0.053	0.795 ± 0.008
Fructose	0.828 ± 0.052	0.679 ± 0.060	0.675 ± 0.023	0.654 ± 0.033	0.784 ± 0.062	0.781 ± 0.037
Glucose	0.772 ± 0.009	0.628 ± 0.064	0.584 ± 0.032	0.700 ± 0.034	0.730 ± 0.023	0.778 ± 0.030
Maltose	0.775 ± 0.028	X	0.604 ± 0.017	0.621 ± 0.016	X	0.804 ± 0.033
Mannose	0.626 ± 0.074	X	X	0.762 ± 0.017	X	0.771 ± 0.046
Xylose	0.841 ± 0.043	0.692 ± 0.038	0.687 ± 0.026	0.691 ± 0.035	0.677 ± 0.036	0.725 ± 0.090
Ø	0.772 ± 0.083	0.675 ± 0.059	0.629 ± 0.05	0.681 ± 0.055	0.713 ± 0.071	0.776 ± 0.054
Lactose	0.472 ± 0.017	0.385 ± 0.022	0.423 ± 0.011	0.432 ± 0.029	0.394 ± 0.097	0.805 ± 0.023
Lactose*	0.966 ± 0.027	0.828 ± 0.044	0.889 ± 0.017	0.885 ± 0.039	0.699 ± 0.152	0.841 ± 0.027

## Data Availability

The data presented in this study are available in the main article or the corresponding Appendix A.

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
