# Peer review of "Isolation and Biochemical Characterization of Six Anaerobic Fungal Strains from Zoo Animal Feces"

_microorganisms, 2021, doi:10.3390/microorganisms9081655_

Round 1
Reviewer 1 Report
The paper describes well the isolation, identification, and the biochemical characterization of anaerobic fungi from zoo animal feces. Anaerobic fungi are microorganisms with good potential that can be used in an integrated biorefinery concept. The abilities of 6 strains of anaerobic fungi grown on monosaccharides are reported in this study glucose-based, lactose as well as wheat straw, a complex substrate on which the attention of biotechnological research is focusing. Although the experiment was conducted in an adequate and rigorous way, minor revisions before paper publication are required:
Abstract
Pag.1 Line 10 Please, remove “Here”, start the phrase with “The isolation…”
P1 L12 Please, add a comma before “is described”.
P1 L14 Please, replace “we” with “was”.
P1 L16 Please, remove “thereby”, replace the full stop with a semicolon and write “hydrogen” instead “Hydrogen”; add a comma after “lactate” before “and”.
P1 L17 Please, remove “Hovewer”.
P1 L20 Please, replace “and the complex substrate” with “or on the straw,”
P1 L20 Please, remove “Therefore”, start the phrase with “Possible”
Introduction
P1 L38 Please, replace “AF” with “anaerobic fungi” at the start of the periods it is better to write in full form. The same at L56.
P2 L58 Please, write “occurs” instead “occur”.
P2 L64 Please, replace “we” with “is” and “report” with “reported”
P2 L66 Please, add a full stop after “Khoyollomyces”.
P2 L66-67 Please rephrase as follows: “The biochemical characterization of these isolates as well as of isolate A252, which was previously identified [19], was also performed.
Materials and methods
Liter and its submultiples should be identified with “L” and not with “l”; therefore, you should correct “l” in “L” and “ml” in “mL” throughout the section.
P2 L70 Please, replace “by” with “previously” which will be written before “described”
P2 L89 Please, rewrite better K2HPO4: it seems that “O” is a zero.
P3 L121 Be careful, watussi cattle should be identified with Bos primigenius f. taurus.
P3 L141-142 Please, adjust the text color with black.
P4 L178 Please, at the start of the phrase replace “5 ml” with “Five mL”.
Results
P5 L201 Be careful, watussi cattle should be identified with Bos primigenius f. taurus.
P7 L241-242 Please, rewrite in impersonal form as follows: “The substrate utilization range by isolates as well as Aestipascuomyces dubliciliberans strain A252, was analyzed and evaluated by quantification of the produced metabolites”.
P7 L244 Please, evaluate the possibility to replace “consistently “with “carefully”. It seems to be more appropriate.
P7 L270 Please, add “and will not be discussed further” after “artefact”.
P7 L273 Please, remove “trehalose” because it has been written that the production of metabolites from trehalose is to be considered as an artefact and it should be not discussed.
P8 L298 Please, replace “the production of SA222” with “for SA222”
P9 L311 Please, replace “secreted” with “produced and replace “at high total metabolite concentrations” with “when an abundant production of other metabolites occurred (Fig. 2), although it remained a marginal metabolite (Fig. 3)
P9 L329-330 Please, evaluate the possibility to replace the full stop before “Although” with a semicolon, add “more” between “was” and “elevated”. Add “than on monosaccharides” after “straw” and replace “compared to” with “vs.”
P9 L333 Please, rewrite as follows: “Metabolite production by strain W212 and G341 followed a similar pattern”
P9 L334 Please, rephrase as follows: “Hydrogen increased significantly while growing on straw (0.375 mmol) in comparison to monosaccharides (0.097-0.194 mmol)” .
P9 L335 Please, replace “compared to” with “vs.”
P9 L357 Please, add a comma after “On straw”.
P9 L359 Please, remove “While” and add “whereas” before “it”
P9 L362 Please, replace “also” with “the”.
Discussion
P13 L380 Please, replace “We isolated” with “In this study,” and add “were isolated” afer “zoo animals” at L381.
P13 L405 Please, replace “Consistent” with “Accordingly”
P13 L417 Please, replace “had” with “used”
P13 L430 Please, add “usage” between “substrate” and “range”.
P14 L448 Please, replace “Instead” with “On the contrary”
P15 L500 Please, replace “while” with “when”.
Author Response
Dear reviewer 1,
first, thank you for your time and effort to read our manuscript carefully. We appreciate your comments which helped us to improve our work. Concerning your remarks:
Abstract
Pag.1 Line 10 Please, remove “Here”, start the phrase with “The isolation…”
Answer: “Here” was removed accordingly.
P1 L12 Please, add a comma before “is described”.
Answer: A comma was added accordingly.
P1 L14 Please, replace “we” with “was”.
Answer: “We” was replaced accordingly.
P1 L16 Please, remove “thereby”, replace the full stop with a semicolon and write “hydrogen” instead “Hydrogen”; add a comma after “lactate” before “and”.
Answer: The sentence was revised accordingly.
P1 L17 Please, remove “Hovewer”.
Answer: “However” was removed.
P1 L20 Please, replace “and the complex substrate” with “or on the straw,”
Answer: The phrases were replaced accordingly.
P1 L20 Please, remove “Therefore”, start the phrase with “Possible”
Answer: The sentence was revised accordingly.
Introduction
P1 L38 Please, replace “AF” with “anaerobic fungi” at the start of the periods it is better to write in full form. The same at L56.
Answer: The phrases were replaced accordingly.
P2 L58 Please, write “occurs” instead “occur”.
Answer: The sentence was revised accordingly.
P2 L64 Please, replace “we” with “is” and “report” with “reported”
Answer: The sentence was revised accordingly.
P2 L66 Please, add a full stop after “Khoyollomyces”.
Answer: The sentence was revised accordingly.
P2 L66-67 Please rephrase as follows: “The biochemical characterization of these isolates as well as of isolate A252, which was previously identified [19], was also performed.
Answer: The sentence was revised accordingly.
Materials and methods
Liter and its submultiples should be identified with “L” and not with “l”; therefore, you should correct “l” in “L” and “ml” in “mL” throughout the section.
Answer: All variations of liter have been change to “L”.
P2 L70 Please, replace “by” with “previously” which will be written before “described”
Answer: The sentence was revised accordingly.
P2 L89 Please, rewrite better K2HPO4: it seems that “O” is a zero.
Answer: We apologize for the mistake, it was corrected accordingly.
P3 L121 Be careful, watussi cattle should be identified with Bos primigenius f. taurus.
Answer: We have improved the species name.
P3 L141-142 Please, adjust the text color with black.
Answer: The text color was adjusted accordingly.
P4 L178 Please, at the start of the phrase replace “5 ml” with “Five mL”.
Answer: The sentence was revised accordingly.
Results
P5 L201 Be careful, watussi cattle should be identified with Bos primigenius f. taurus.
Answer: We have improved the species name.
P7 L241-242 Please, rewrite in impersonal form as follows: “The substrate utilization range by isolates as well as Aestipascuomyces dubliciliberans strain A252, was analyzed and evaluated by quantification of the produced metabolites”.
Answer: The sentence was revised accordingly.
P7 L244 Please, evaluate the possibility to replace “consistently “with “carefully”. It seems to be more appropriate.
Answer: We had another look on the sentence. While we shared your opinion of “consistently” not being completely appropriate, we think that “reliably” makes more sense in this place.
P7 L270 Please, add “and will not be discussed further” after “artefact”.
Answer: The phrase was added accordingly.
P7 L273 Please, remove “trehalose” because it has been written that the production of metabolites from trehalose is to be considered as an artefact and it should be not discussed.
Answer: “Trehalose” was removed accordingly.
P8 L298 Please, replace “the production of SA222” with “for SA222”
Answer: “Of” was replaced with “for”.
P9 L311 Please, replace “secreted” with “produced and replace “at high total metabolite concentrations” with “when an abundant production of other metabolites occurred (Fig. 2), although it remained a marginal metabolite (Fig. 3)
Answer: Both phrases have been replaced accordingly.
P9 L329-330 Please, evaluate the possibility to replace the full stop before “Although” with a semicolon, add “more” between “was” and “elevated”. Add “than on monosaccharides” after “straw” and replace “compared to” with “vs.”
Answer: We think that by replacing the full stop the sentence is getting to long worsening the understandability. Your other suggestions were adopted.
P9 L333 Please, rewrite as follows: “Metabolite production by strain W212 and G341 followed a similar pattern”
Answer: The sentence was revised accordingly.
P9 L334 Please, rephrase as follows: “Hydrogen increased significantly while growing on straw (0.375 mmol) in comparison to monosaccharides (0.097-0.194 mmol)”.
Answer: We adopted your suggestion. Also, we added “being produced” at the end of the sentence.
P9 L335 Please, replace “compared to” with “vs.”
Answer: The sentence was revised accordingly.
P9 L357 Please, add a comma after “On straw”.
Answer: The sentence was revised accordingly.
P9 L359 Please, remove “While” and add “whereas” before “it”
Answer: The sentence was revised accordingly.
P9 L362 Please, replace “also” with “the”.
Answer: Also” was replaced with “the”.
Discussion
P13 L380 Please, replace “We isolated” with “In this study,” and add “were isolated” afer “zoo animals” at L381.
Answer: The sentence was revised accordingly.
P13 L405 Please, replace “Consistent” with “Accordingly”
Answer: The sentence was revised accordingly.
P13 L417 Please, replace “had” with “used”
Answer: The sentence was revised accordingly.
P13 L430 Please, add “usage” between “substrate” and “range”.
Answer: The sentence was revised accordingly.
P14 L448 Please, replace “Instead” with “On the contrary”
Answer: The sentence was revised accordingly.
P15 L500 Please, replace “while” with “when”.
Answer: The sentence was revised accordingly.
Reviewer 2 Report
In this manuscript, authors performed isolation of five and characterization of six new anaerobic fungal strains from zoo animal feces. In general I find this work very interesting for scientific community focused on anaerobic fungi. The manuscript is well organised and well written and give inquiring perspectives for future studies. In my opinion this work has enough scientific merit for publication in Microorganisms. However, manuscript body needs to be improved and some issues needs to be clarified before final manuscript acceptance.
Some major problems that should be addressed by the Authors are discussed below:
- Introduction Section: lines 31-42: The text fragment needs to be rewritten. In my opinion, despite the great advantages of the biotechnological use of anaerobic fungi and their enzymes, they have several limitations especially in the context of lignocellulose degradation. E.g.: one of the best known enzyme able to breaking down lignin is laccase, which needs oxygen as a final electron acceptor. In an anaerobic environment, the enzyme is unable to properly and efficiently catalyze the reaction.
- Line 68: there should be should a headline: materials and methods
- Line 78: Please specify in the manuscript body the name and manufacturer of a milling apparatus.
- Line 127: Please describe in the manuscript body the deoxygenation method or apparatus.
- Line 129: Please specify in the manuscript body where, what apparatus was used for fungi cultivation.
- Line 132, 134 and 136: Please describe in the manuscript body how the inoculation was performed?
- Line 140: how many grams of mycelium was used for DNA isolation? Please supplement the information
- Line 148: what was the quantity of DNA (in mg not μl) used in PCR reaction? Please supplement the information.
- Line 200-203: Is it not surprising that only one fungal strain has been isolated from each sample? Please give reason for this and discuss in the Discussion Section.
- Line 250-274 and Table 1: this manuscript fragment and Table 1 present the same results. It is a repetition of the results and the text fragment should be rewritten.
- Figure 2: Error bars in Fig 2 suggest that same statistical analysis was performed. I was not able to find such information in materials and methods section. Please provide information on statistical analysis preformed? Were the measurements performed in duplicate? Triplicate?
Minor suggestions:
- Line 44-45: sentence needs to be rewritten
- Line 486, 487, 488: repetition: 'amount of'
Therefore, the reviewer suggests MINOR manuscript correction.
Author Response
Dear reviewer 2,
first, thank you for your time and effort to read our manuscript carefully. We appreciate your comments which helped us to improve our work. Concerning your remarks:
- Introduction Section: lines 31-42: The text fragment needs to be rewritten. In my opinion, despite the great advantages of the biotechnological use of anaerobic fungi and their enzymes, they have several limitations especially in the context of lignocellulose degradation. E.g.: one of the best known enzyme able to breaking down lignin is laccase, which needs oxygen as a final electron acceptor. In an anaerobic environment, the enzyme is unable to properly and efficiently catalyze the reaction.
Answer: We have improved the section by specifying the limitation of AF for lignin degradation.
- Line 68: there should be should a headline: materials and methods
Answer: We changed the headline from “Methods” to “Materials and Methods” as you suggested.
- Line 78: Please specify in the manuscript body the name and manufacturer of a milling apparatus.
Answer: We added the name and manufacturer of the cutting mill.
- Line 127: Please describe in the manuscript body the deoxygenation method or apparatus.
Answer: We specified the deoxygenation procedure.
- Line 129: Please specify in the manuscript body where, what apparatus was used for fungi cultivation.
Answer: We added specific information for the apparatus to the manuscript.
- Line 132, 134 and 136: Please describe in the manuscript body how the inoculation was performed?
Answer: The information was added accordingly.
- Line 140: how many grams of mycelium was used for DNA isolation? Please supplement the information
Answer: We used the entirety of a one-week old culture grown as described in text. Weighting of the mycelium would have yielded in a high loss of mycelium. We edited the text to make it clear that the whole mycelium from a culture was used for the extraction.
- Line 148: what was the quantity of DNA (in mg not μl) used in PCR reaction? Please supplement the information.
Answer: We added the quantity of DNA to the text.
- Line 200-203: Is it not surprising that only one fungal strain has been isolated from each sample? Please give reason for this and discuss in the Discussion Section.
Answer: It is true that the presence of many species has been confirmed in a single feces sample by applying molecular methods. However, it has also been reported that special efforts have to be undertaken to isolate more than the most abundant species from the same sample. Since we didn´t aim to explore fungal diversity but rather to isolate the more robust strains from the samples, we have not undertaken special efforts and therefore, we most likely isolated only the most abundant species from each sample. In this regard it is not surprising that we only isolated one species per sample because it can be attributed to our isolation routine. We added a paragraph discussing this at the end of the “Phylogeny” chapter of the discussion.
- Line 250-274 and Table 1: this manuscript fragment and Table 1 present the same results. It is a repetition of the results and the text fragment should be rewritten.
Answer: We humbly disagree. The text evaluates the growth of the different isolates by ranking the growth in good, low and no growth. The table expands this information by allowing a comparison of the total produced metabolites, thereby further differentiating the growth capabilities of the fungi. Adding all numbers to the text would, in our opinion, impair readability.
- Figure 2: Error bars in Fig 2 suggest that same statistical analysis was performed. I was not able to find such information in materials and methods section. Please provide information on statistical analysis preformed? Were the measurements performed in duplicate? Triplicate?
Answer: All experiments were performed in triplicates. All given numbers are the mean value ± standard deviation. Additional statistical analyses were not conducted. We added that the experiments were performed in triplicate to the materials and methods section (subsection Carbon source usage and analytics) and in the figure captions to clarify.
Minor suggestions:
- Line 44-45: sentence needs to be rewritten
Answer: The sentence has been rewritten.
- Line 486, 487, 488: repetition: 'amount of'
Answer: We changed the phrase to avoid repetition.